# Enhancement of the Local CD8^+^ T-Cellular Immune Response to *Mycobacterium tuberculosis* in BCG-Primed Mice after Intranasal Administration of Influenza Vector Vaccine Carrying TB10.4 and HspX Antigens

**DOI:** 10.3390/vaccines9111273

**Published:** 2021-11-03

**Authors:** Kirill Vasilyev, Anna-Polina Shurygina, Natalia Zabolotnykh, Mariia Sergeeva, Ekaterina Romanovskaya-Romanko, Anastasia Pulkina, Janna Buzitskaya, Marine Z. Dogonadze, Tatiana I. Vinogradova, Marina A. Stukova

**Affiliations:** 1Smorodintsev Research Institute of Influenza of the Ministry of Health of the Russian Federation, 197376 St. Petersburg, Russia; ann-polin@yandex.ru (A.-P.S.); mari.v.sergeeva@gmail.com (M.S.); romromka@yandex.ru (E.R.-R.); pureska@mail.ru (A.P.); janna.buzitskaya@influenza.spb.ru (J.B.); marina.stukova@influenza.spb.ru (M.A.S.); 2Saint-Petersburg State Research Institute of Phthisiopulmonology of the Ministry of Health of the Russian Federation, 191036 St. Petersburg, Russia; zabol-natal@yandex.ru (N.Z.); marine-md@mail.ru (M.Z.D.); vinogradova@spbniif.ru (T.I.V.)

**Keywords:** *M. tuberculosis* vaccine, TB10.4, HspX, influenza vector, mucosal immunization, innate immune response, T-cellular immune response, flow cytometry

## Abstract

BCG is the only licensed vaccine against *Mycobacterium tuberculosis (M.tb)* infection. Due to its intramuscular administration route, BCG is unable to induce a local protective immune response in the respiratory system. Moreover, BCG has a diminished ability to induce long-lived memory T-cells which are indispensable for antituberculosis protection. Recently we described the protective efficacy of new mucosal TB vaccine candidate based on recombinant attenuated influenza vector (Flu/THSP) co-expressing TB10.4 and HspX proteins of *M.tb* within an NS1 influenza protein open reading frame. In the present work, the innate and adaptive immune response to immunization with the Flu/THSP and the immunological properties of vaccine candidate in the BCG-prime → Flu/THSP vector boost vaccination scheme are studied in mice. It was shown that the mucosal administration of Flu/THSP induces the incoming of interstitial macrophages in the lung tissue and stimulates the expression of co-stimulatory CD86 and CD83 molecules on antigen-presenting cells. The T-cellular immune response to Flu/THSP vector was mediated predominantly by the IFNγ-producing CD8^+^ lymphocytes. BCG-prime → Flu/THSP vector boost immunization scheme was shown to protect mice from severe lung injury caused by *M.tb* infection due to the enhanced T-cellular immune response, mediated by antigen-specific effector and central memory CD4^+^ and CD8^+^ T-lymphocytes.

## 1. Introduction

According to the WHO global tuberculosis report, TB remains one of the top 10 causes of death worldwide. A great variety of tuberculosis vaccine candidates is at different stages of clinical efficacy trials, including whole cell-derived vaccines (killed or attenuated through genetic modifications), subunit and recombinant viral-vectored vaccines (reviewed in [1,2,3]). Currently, the only approved TB vaccine, Bacille Calmette–Guérin (BCG) is widely used with the coverage of about 90% in 120 countries providing BCG vaccination (WHO, Geneva, Switzerland, 2018). However, the effectiveness of BCG in preventing the disease is restricted by early childhood and does not exceed 50% (WHO, Geneva, Switzerland, 2013). Improving the techniques of specific tuberculosis prevention remains one of the priorities of phthisiology.

The low effectiveness of BCG in providing the control of latent endogenous infection or exogenous reinfection in adults could be explained by the insufficient generation of specific long-lived memory T-cells after BCG administration [4]. Moreover, several studies have shown that BCG is ineffective in CD8^+^ T-cell induction [5,6] and cannot provide an adequate immune response in patients with immunocompromised or Th-2 polarized immune states, including HIV-infection and parasitic invasions [7]. Therefore, an apparent failure of BCG to induce significant numbers of central memory T (Tcm) cells may be an important factor to its limited efficacy [8]. Because of its intradermal route of administration, BCG has a disadvantage in comparison with mucosal-administered vaccines which can induce a local immune response at the mucous membranes of the upper respiratory tract [9]. The majority of TB vaccine candidates which a now evaluated in human clinical trials are also only suitable for the parenteral route of delivery. However, mucosal immunization provides better protection from *M. tuberculosis,* entering through the respiratory tract (reviewed in [10]). Among the existing vaccine candidates, only two viral vectors utilize intranasal and aerosol administration route (TB/FLU-04L (RIBSP, Gvardeyskiy, Kazakhstan, NCT02501421) and ChAdOx185A (University of Oxford, Oxford, UK, NCT04121494)).

In the present work, we investigate the innate and adaptive immune response in mice immunized with recombinant influenza A/PR/8/34 strain, expressing a hybrid molecule consisting of mycobacterial antigens TB.10.4 and HspX. Previously it was shown that TB10.4 serves as a target for antimycobacterial immune responses in BCG-immunized humans and mice. More than 70% of vaccinated or TB-infected individuals react on restimulation with TB10.4 by high levels of IFN-γ production [11]. HspX demonstrates an immunoadjuvant potential, inducing DC maturation and proinflammatory cytokine expression through the activation of MyD88 and TRIF signaling pathways [12]. It can induce strong cellular and humoral immune responses in latent phase of tuberculosis infection [13,14,15,16]. The perspectives of the use of HspX protein as a candidate vaccine against *Mycobacterium tuberculosis* are discussed in the review by Yousefi-Avarvand et al. [17]. It should be also noted that both TB10.4 and HspX are encoded by Bacillus Calmette-Guerin which makes them useful in a prime-boost immunization scheme [18].

The sequence encoding TB.10.4 and HspX proteins was inserted in the NS1 ORF of the virus after the first 124 amino acid codons. The truncation of NS1 decreases its antagonistic influence on the IFN-I signaling system and leads to the inhibition of viral replicative activity in the respiratory tract of humans and laboratory animals [19,20,21,22]. At the same time, NS1-truncated mutants provoke a strong innate immune reaction driving the formation of pronounced Th-1 polarized adaptive immune response, accompanied by the generation of polyfunctional effector CD8^+^ T-lymphocytes [22,23,24,25,26,27]. High immunogenicity and self-adjuvant properties of influenza vectors based on NS1-modified strains make them a perfect tool for the delivery of foreign antigens into the organism for the induction of immune response in the site of primary respiratory infection. The protective efficacy of ESAT-6 expressing influenza vector against Mycobacterium tuberculosis (*M.tb*) was shown previously [25,28]. Here we study the innate immune cells population dynamics, the expression of activation markers of the immune system and the generation of effector and memory TB10.4- and HspX-specific T-cells after the mucosal administration of A/PR8/NS124-TB10.4-2A-HspX influenza strain. Further, we analyze the T-cellular immune response to our vaccine candidate in heterologous BCG-prime → Flu/THSP vector boost immunization protocol on the model of experimental TB infection with the virulent *M.tb* strain H37Rv.

## 2. Materials and Methods

### 2.1. Animals

C57/black-6 6–8 weeks old male mice were obtained from the Biomedical Science Center (Stolbovaya, Moscow, Russia). All animal studies were done in accordance with the international recommendations (Directive 2010/63/EU), and the protocols approved by the Bioethic Committee at the Smorodintsev Research Institute of Influenza and by the Bioethic Committee at the Saint-Petersburg State Research Institute of Phthisiopulmonology.

### 2.2. Vaccine Strain 

Chimeric influenza vaccine virus construction Flu/THSP was generated by reverse genetic as described [29]. Virus infectious activity was determined by titration in CE, Vero (ATCC), or MDCK cells (IRR, #FR-58). Infection was detected by positive hemagglutination and the 50% infectious dose was calculated by the method of Reed-Muench. 

### 2.3. Experimental Design 

Mice were intranasally immunized with 30 µL of Flu/THSP vaccine virus (6.0 lg EID_50_/animal) or an equivalent volume of DPBS (control group). In order to analyze the dynamics of innate immune cell populations, animals were sacrificed 24, 48 and 78 h post single intranasal Flu/THSP immunization and lungs were collected. Analysis of the adaptive T-cell immune response following a single intranasal Flu/THSP vector administration was performed in lungs and spleens 8 days post immunization.

For prime-boost immunization mice were first subcutaneously primed with 100 µL of BCG vaccine (5.0 lg CFU/animal). 24 weeks later an intranasal boost immunization with 30 µL of Flu/THSP vaccine virus (6.0 lg EID50/animal) was performed. BCG-only group and control animals received the equivalent volume of DPBS. Six weeks after the boost vaccination, all animals were inoculated via the lateral tail vein with 6.0 lg CFU/animal of virulent *M. tuberculosis* H37Rv strain suspended in 200 µL of DPBS. Five weeks post *M. tuberculosis* challenge lungs and spleen were collected for analysis of the adaptive T-cell immune response.

### 2.4. Cell Preparation

Mice were sacrificed, chest was opened and right ventricle was perfused with 10 mL of ice-cold DPBS (Biolot, St. Petersburg, Russia) before the removal of lungs and spleen. Following the mechanical dissociation of spleen and collagenase/DNAse (Sigma, Saint Louis, MO, USA) digestion of lung tissue, cells were filtered through 70 μm cell strainer. Erythrocytes were lysed with RBC lysis buffer (BioLegend, San Diego, CA, USA) in accordance with the manufacturer protocol. Cells were seeded at a density 10^6^ cells per well into flat bottom 96-well tissue culture plates (Nunc, Roskilde, Denmark)) in RPMI 1640 (Gibco) medium supplemented with 10% *v*/*v* FBS (Gibco, Waltham, MA, USA) and 1% penicillin-streptomycin solution (Gibco). 

For adaptive T-cell immune response analysis cells were stimulated with the mixture of recombinant TB10.4 (Research Institute of Influenza, St. Petersburg, Russia) or HspX (MyBiosourse, San Diego, CA, USA) proteins in final concentration of 5 μg/mL in a presence of anti-CD28 antibodies (BioLegend) for 24 h at 37 °C, 5% CO_2_. The GolgiPlug (BD Biosciences, San Jose, CA, USA) reagent was added 6 h before the end of stimulation. Medium alone and PMA/Ionomycin mix (Sigma) were used as negative and positive controls respectively.

### 2.5. Flow Cytometry Analysis

Two panels of fluorochrome-conjugated antibodies (BioLegend, San Diego, CA, USA) were used to identify the innate immunity cell populations and to analyze the activation marker expression: (1) CD11b-PE/Cy7, CD11c-PE, MHCII-Alexa488, CD103-PerCP-Cy5.5, CD45-APC/Cy7, CD64-BV421, CD24-BV510; (2) CD45-APC/Cy7, MHCII-Alexa488, Ly6G-PerCP-Cy5.5, CD86-BV421, CD83-BV510. T-cells staining was performed using the fluorochrome-conjugated antibody set containing CD4-PerCP-Cy5.5, CD8-PE/Cy7, CD62L-APC/Cy7, and CD44-BV421 (BioLegend, San Diego, CA, USA). Intracellular production of cytokines was assessed using antibodies against IFNγ-FITC, IL2-PE, and TNFα-BV510 (BioLegend, San Diego, CA, USA). Staining for the detection of intracellular markers was performed using the Fixation and Permeabilization Solution reagent kit (BD Biosciences, San Jose, CA, USA) according to the manufacturer’s instructions. Zombie Red viability marker (BioLegend, San Diego, CA, USA) was used to identify the dead cells. True Stain reagent (BioLegend, San Diego, CA, USA), containing antibodies to CD16/CD32, was added during the surface markers staining to block non-specific antibody binding. Data were collected on a Cytoflex flow cytometer (Beckman Coulter, Bray, CA, USA). The results were analyzed using the Kaluza Analysis 2.2 program (Beckman Coulter, Bray, CA, USA).

### 2.6. Protection Study in Mice

The severity of experimental tuberculosis was assessed 5 weeks after challenge by evaluating bacterial load in lungs and spleen, mass coefficients of lungs and spleen, lung damage score and histological evaluation of inflammation process in lungs. Based on the number and size of TB specific lesions and presence the areas of necrosis in the lungs, gross pathological scores were graded from 1 to 4 conventional units according to the following criteria: scanty small tubercles were estimated at 0.5 units (U), small tubercles (<5) as 1.0 U, numerous small tubercles (>5) as 1.5 U, occasional large tubercles as 1.75 U, confluent small tubercles and occasional large tubercles as 2.0 U, large tubercles (<10) as 2.25 U, numerous large tubercles (>10) as 2.5 U, numerous confluent tubercles as 2.75 U, tubercles with areas of necrosis as 3.0 U, numerous large necrotic tubercles as 4.0 U. In case of lung maceration by serous liquid, the index was increased by 0.25 to 1.0 U, depending on the extent of damage.

Phagocytosis was studied in the culture of peritoneal macrophages obtained from the lavage of the abdominal cavity of mice in relation to the cell suspension of the yeast Saccharomyces cerevisiae opsonized with mouse serum. Phagocytic activity was measured microscopically as the amount of yeast cells digested by macrophages after 1.5 h of cultivation.

### 2.7. Mycobacterial Load

For the quantification of live mycobacteria load, tissue homogenates were titrated and cultured on solid Lowenstein-Jensen medium. Bacterial colonies were counted after 3 weeks of incubation at 37 °C. Titers were expressed as log10 of the mean colony forming units (lg CFU) per lung weight. The detection limit was equal to 2 × 10^3^ CFU. Decrease in bacterial load more than 0.5 lg CFU in comparison to the control group was considered as a positive protective effect.

### 2.8. Histopathology

Lung tissues were fixed in 10% formalin (pH 7.0), followed by paraffin embedment. For histopathological studies, 5–6 mm sections were stained with hematoxylin and eosin. Images were captured using Olympus BX45 microscope (Olympus Corp., Tokyo, Japan) with camera and Olympus DP-Soft software package (Olympus Corp., Tokyo, Japan).

### 2.9. Statistical Analysis

Statistical processing of the results was carried out using the RStudio Desktop 1.0.153 program (RStudio Inc, Boston, MA, USA). The experimental and control groups were compared using the Dunnett test. The comparison of the experimental groups was carried out using the Student *t*-test.

## 3. Results

### 3.1. Flu/THSP Recombinant Virus Efficiently Activates Innate Immune Response at the Site of Administration 

A panel of fluorescently-labelled antibodies to CD45, MHCII, CD11c, CD11b, CD24, CD64 and Ly6G [27,30,31] was used to study the innate immune cell populations in the lungs in response to intranasal administration of the Flu/THSP vaccine strain. This allowed distinguishing neutrophils, monocytes, alveolar and interstitial macrophages, as well as CD11b^−^ and CD11b^+^ dendritic cells. The gating strategy is presented in Appendix A.

Innate immune cell population analysis after the immunization revealed that Flu/THSP vector caused a significant increase in the interstitial macrophage population percentage at all studied time points (24 h: *p* = 0.0001; 48 h: *p* = 0.001; 72 h *p* = 0.01). The percentage of alveolar macrophage population in immunized group was higher compared to control at 24 h post-immunization (*p* = 0.001), but no difference was observed at later time points. The increase in the macrophage level in the lungs was accompanied by a significant reduction in the CD11^−^ and CD11^+^ dendritic cells (DC) content, especially at 48 and 72 h after vaccine administration. The relative number of monocytes and neutrophils was also higher in the Flu/THSP -treated group, but the difference from the control was less pronounced (Figure 1A). Thus, intranasal Flu/THSP immunization induced an attraction of macrophages and neutrophils in the lung tissue and the egress of CD11b^−^ and CD11b^+^ dendritic cells. In the vaccine-treated group the predominance of interstitial macrophages was observed, while in the control group neutrophils were the dominant population among the investigated cell types at 24 and 48 h after the treatment (Figure 1B). 

To investigate the capacity of Flu/THSP vaccine candidate to stimulate the antigen-presenting cells (APC) we analyzed the expression of activation markers CD83 and CD86 on CD45^+^MHCII^+^ cells. The results of the analysis of the median fluorescence intensity (MFI) of the activation markers CD83 and CD86 on the APC (CD45^+^MHCII^+^) are presented in Figure 1C. It was shown that immunization leads to the enhancement of costimulatory CD86 molecule expression on each time point. The same results were obtained for the CD83 activation marker.

### 3.2. Single Flu/THSP Immunization Induces Both Local and Systemic Antigen-Specific T-Cell Response

Antigen-specific T-cell immune response was studied in lungs and spleens on 8th day post immunization. Effector and central memory CD4^+^/CD8^+^ T-cell subsets, expressing IFNγ, TNFα and IL-2 were studied following 24 h stimulation with the mixture of recombinant TB10.4 and HspX proteins in ICS assay. 

Flu/THSP vaccinated animals showed a significant increase in the relative number of cytokine-producing CD4^+^ and CD8^+^ Tem cells both in the lungs and spleens. In the lungs, the CD4^+^ Tem response to stimulation was mainly represented by TNFα^+^ cells, whereas in the spleens, IFNγ single-positive cells dominated the population of CD4^+^ Tem. 

The repertoire of CD8^+^ Tem cells in response to stimulation was characterized by a predominant increase in the relative number of IFNγ single-positive cells. However, the double positive populations of IFNγ^+^TNFα^+^ T-lymphocytes were presented in the lungs and IFNγ^+^IL-2^+^, IFNγ^+^TNFα^+^ and IL-2^+^TNFα^+^ in the spleens (Figure 2). There was no statistically significant increase in the total level of cytokine-producing Tcm cells in response to protein stimulation. 

### 3.3. Prime-Boost Immunization with Flu/THSP Induces Better Protection against M. tb Strain H37Rv Compared to Single BCG Immunization

The prime-boost vaccination effectiveness was measured in mouse model of TB infection 5 weeks after the challenge with *M. tuberculosis* H37Rv strain. The most prominent protective effect, observed in BCG prime → Flu/THSP boost group, was associated with the significant decrease in bacterial loads in lungs and spleen compared with control (mean decrease by 1.23 lg CFU and 1.02 lg CFU, respectively) and single BCG-vaccinated groups (mean decrease by 0.62 lg CFU and 0.5 lg CFU in lungs and spleen, respectively). Lung damage score, as well as mass coefficients of organs were also lower in Flu/THSP boost group. The measurement of phagocytosis activity of peripheral macrophages (PMF) showed that the BCG-prime → Flu/THSP boost immunization scheme prevented the inhibition of absorption and digestion capacity of PMF, generally observed in the disseminated tuberculosis infection in mice (Figure 3). 

The histopathological investigation showed the most prominent lung injury in the control (non-vaccinated) group. Confluent foci of specific infiltration occupied more than 30% of the section area. Lymphocytes, macrophages and epithelioid cells were presented in the infiltrate as well as single neutrophils and their accumulations (Figure 4a,b). 

In mice primed with BCG and boosted with Flu/THSP only minor inflammatory reactions were observed. No confluent infiltration was shown and the specific inflammation foci were small, lack exudate and had a perivascular or peribronchial localization. All granulomas in the prime-boost immunized group were lymphoid, while non-vaccinated animals had epithelioid cell granulomas with large epithelioid cell accumulations (Figure 4c,d).

Single BCG immunized mice were characterized by intermediate severity of inflammation. Granulomas with single epithelioid cells in the center were shown in this group (Figure 4e,f).

### 3.4. Prime-Boost Immunization with Flu/THSP Induces Higher Local and Systemic Antigen-Specific T-Cell Response Than Single BCG Vaccination

Further, we compared the T-cellular immune response to BCG-prime → Flu/THSP boost vaccination with single BCG immunization in the context of TB infection.

A significant increase in the proportion of cytokine-producing CD4^+^ Tem (CD44^+^CD62L^−^) in spleens after the protein stimulation was observed in all vaccinated groups of animals compared to infection control (Figure 5). In the BCG vaccinated group the level of TNFα^+^-single-positive cells were higher compared to control group (*p* = 0.03), while the Flu/THSP-boosted group demonstrated an augmented formation of IFNγ^+^TNFα^+^-bifunctional (*p* = 0.002) and IFNγ^+^-monofunctional (*p* = 0.02) CD4^+^ Tem cells. At the same time, Flu vector-boosted mice had an elevated number of IFNγ^+^TNFα^+^ and IFNγ^+^ CD8^+^ Tem both in lungs and spleen showing the enhanced local and systemic *M. tb.*-specific T-cell response. In the BCG-treated group the level of discussed populations was lower or the same as in the control group (Figure 5). 

The difference between the studied groups in the percentage of cytokine-secreting CD4^+^ and CD8^+^ central memory T-cells (CD44^+^CD62L^+^) was more profound. As shown on Figure 5, BCG-vaccinated animals demonstrated the lowest level of TB10.4/HspX-specific CD8^+^ Tcm response in lungs, while in the spleen the amount of cytokine-producing Tcm was higher than in control group (Figure 6). Flu/THSP -boosted mice had the highest level of Tcm immune response (CD4^+^ and CD8^+^) both in lungs and spleen. The significant differences between Flu/THSP boost and BCG groups were obtained for CD4/8^+^ IFNγ^+^ single-producers in lungs (*p* = 0.009, 0.001, respectively), for IFNγ^+^TNFα^+^ bifunctional CD8^+^ T-cells in lungs (*p* = 0.02) and for the total amount of cytokine-producing CD8^+^ Tcm in spleen of infected mice. It is noteworthy that in the control group the number of CD4^+^ TNFα-single-producers was higher than in other experimental groups (*p* = 0.02, 0.01).

The stimulation of splenocytes and lung cells of experimental animals with BCG led to similar results as TB10.4/HspX-mix stimulation (Appendix A). However, the total amount of cytokine-producing cells after the 24 h incubation with BCG was 1.5–2 times higher in all groups compared to protein stimulation. Furthermore, BCG stimulation allowed to reveal the populations of IFNγ^+^IL-2^+^TNFα^+^ polyfunctional CD4/8^+^ T-cells. It was shown, that in the vector boosted group the amount CD4^+^ Tem triple cytokine producers was higher than in the BCG and control group (*p* = 0.03).

## 4. Discussion

The protective efficacy of anti-tuberculosis immune response is largely dependent on Th1-polarised CD4^+^ T-cells and CD8^+^ CTLs [32,33,34,35,36]. Thus, the success of anti-TB vaccination is tightly associated with the ability of vaccines to induce functional Th1 and CTL subpopulations. In this regard, attenuated influenza viral vectors seem promising candidates for anti-tuberculosis vaccine formulations due to their ability to stimulate the formation of polyfunctional T-lymphocytes which are indispensable for successful protection against *M. tuberculosis* [37]. 

In our previous work we described the protective efficacy of mucosal influenza vector vaccine carrying TB10.4 and HspX antigens against *M. tuberculosis* on mice and guinea pigs models. The Flu/THSP vector was safe and induced TB specific CD4^+^ and CD8^+^ T-cell immune response after intranasal administration in mice. Protection against two virulent *M. tuberculosis* strains was shown in mice and guinea pigs after intranasal immunization with the Flu/THSP vector. Moreover, it was shown that recombinant influenza vector enhances the protection against *M. tuberculosis* in heterologous BCG-prime → Flu/THSP vector boost immunization scheme compared to BCG immunization alone [38]. 

We investigate the local innate immune reaction to the Flu/THSP vector administration in mouse lungs. It was shown that Flu/THSP induces the incoming of interstitial macrophages and stimulates the expression of co-stimulatory CD86 and CD83 molecules on APCs. The data are in accordance with previously obtained for influenza strain with shortened NS1-protein [26]. CD86 and CD83 expression enhancement is associated with APC maturation and antigen presentation [39,40].

Intranasal immunization with Flu/THSP vector predominantly induced the generation of CD8^+^-Tem response. CD8 T-cells facilitate the elimination of pathogen both by production of cytokines (IL-2, IFNγ, TNFα) and by direct killing of *M. Tuberculosis*-infected cells via apoptosis induction through Fas-Fas-ligand interaction or using perforin and granzymes [41]. We found that the vast majority of antigen-specific cells in the lungs and spleen of Flu/THSP immunized animals were IFNγ single-producers, but bifunctional cells (IFNγ^+^IL-2^+^ and IFNγ^+^TNFα^+^) were also presented. The critical role of IFNγ for the protection from TB infection is well understood and the level of antigen-specific IFNγ response is used as a correlate of TB vaccine effectiveness [42,43]. However, in recent works, much attention is paid to the role of a polyfunctional populations of T-cells in the course of TB-infection. It is shown, that IFNγ^+^TNFα^+^ T cells effectively eliminate the pathogen and can be considered as correlates of protection against *M. tuberculosis* [44,45,46,47]. 

In the experiment with BCG-prime → Flu/THSP vector boost immunization followed by *M. tuberculosis* H37Rv challenge infection we have shown that the T-cellular antigen-specific immune response on restimulation with TB10.4/HspX protein mix was mediated by IFNγ^+^ and IFNγ^+^TNFα^+^ cells (both CD4^+^ and CD8^+^). The highest level of cytokine-producing cells was shown in Flu/THSP boosted animals. Interestingly, the most pronounced difference between BCG and THSP group was in the proportion of antigen-specific central memory T-cells both in the lungs and spleen. Thus the use of intranasal influenza vector carrying TB10.4 and HspX antigens in prime-boost immunization scheme could be a perspective approach to improve protective effects of BCG by activating the innate and adaptive local immune response, enhancing the expansion of antigen specific polyfunctional CD8^+^ T cells and increasing the formation of long-lived central memory T-cells. 

## Figures and Tables

**Figure 1 vaccines-09-01273-f001:**
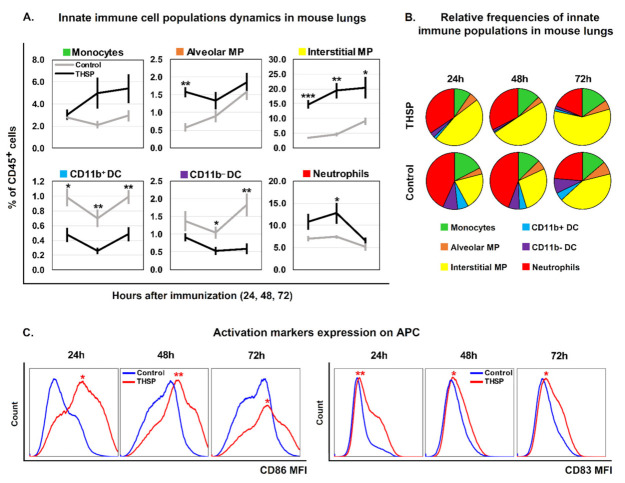
Flow cytometric analysis of innate immune cell populations in mouse lungs after immunization. (**A**) Line graphs depict innate immune cell profile in mouse lungs on 24, 48 and 72 h after intranasal Flu/THSP vaccine strain (THSP) or PBS (Control) administration. Data shown are Means ± SE of corresponding innate immune cell population proportion within the total CD45^+^ subset. Differences between groups were estimated using Student’s t test. * *p* < 0.05, ** *p* < 0.01, and *** *p* < 0.001, *n* = 5. (**B**) Pie charts show the relative frequencies of innate immune cell populations in mouse lungs at indicated time points. (**C**) Flow cytometric analysis of activation markers expression on APC (CD45^+^MHCII^+^). Histograms show the distribution of CD86 and CD83 fluorescence intensity among APC in mouse lungs on 24, 48 and 72 h after intranasal administration of Flu/THSP vaccine strain (THSP) or PBS (Control). Each histogram is a composite of 5 different samples. Median fluorescence intensity (MFI) values of each sample were used for group comparison in Student’s t test. * *p* < 0.05, ** *p* < 0.01, *n* = 5.

**Figure 2 vaccines-09-01273-f002:**
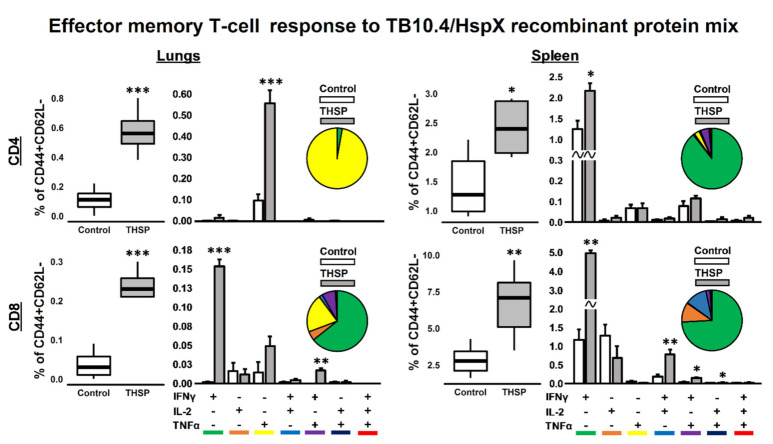
Antigen-specific effector memory T-cellular immune response in mouse lungs and spleen, 8 days after intranasal immunization with Flu/THSP vaccine strain (THSP). CD4/8^+^CD44^+^CD62L^−^ effector memory (EM) T-cells from lungs and spleen expressing IFNγ, IL-2 and TNFα after 24 h stimulation with TB10.4/HspX recombinant protein mix. Box-plots represent total amount of cytokine-producing EM T-cells. Bar-plots represent the average percent of each cytokine-producing population within total EM T-cells subset (Mean ± SE). Pie-charts show the average percent of each cytokine-producing population within total subset of cytokine-producing cells. Differences between groups were analyzed using Student’s t test. * *p* < 0.05, ** *p* < 0.01, and *** *p* < 0.001, *n* = 5.

**Figure 3 vaccines-09-01273-f003:**
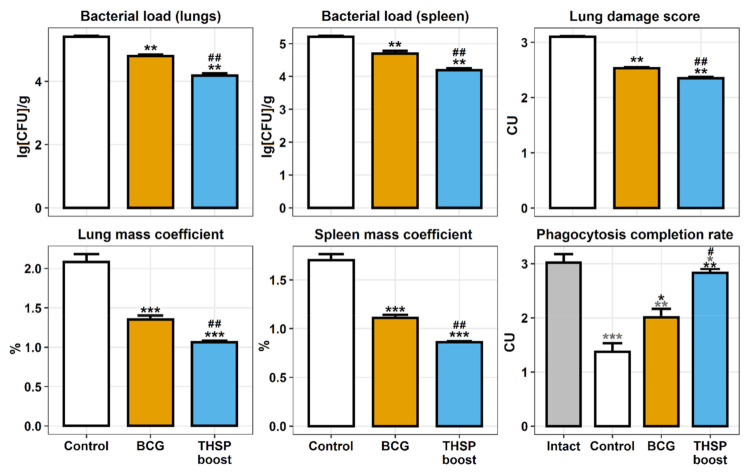
Protective efficacy of BCG-prime → Flu/THSP boost immunization in mouse model of TB infection. Bacterial load in lungs and spleen was measured by titration of tissue homogenates. Lungs and spleen mass coefficients were estimated as the percentage of organ weight relative to body weight. The lung damage score was quantified in conventional units as described in 2.6 section. Phagocytosis completion rate of model yeast cells was measured. Statistical analysis was performed by two-way ANOVA, and the results of the Bonferroni post-hoc test for pairwise comparisons. Black asterisks correspond to control and experimental groups comparisons. Grey asterisks correspond to comparisons with intact group (* *p* < 0.05, **: *p* < 0.01, ***: *p* < 0.001). Sharp signs correspond to BCG and Flu/THSP (THSP) boost group comparisons (# *p* < 0.05, ##: *p* < 0.001).

**Figure 4 vaccines-09-01273-f004:**
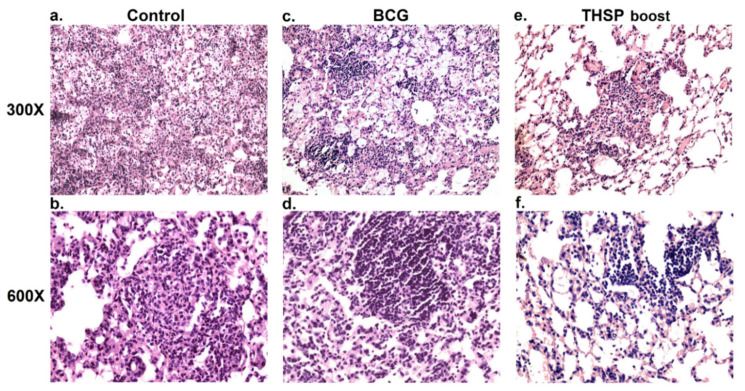
Protective efficacy of BCG-prime Flu/THSP boost immunization strategy in mouse model of TB infection. Representative hematoxylin and eosin (H&E) stained lung histological sections of control (**a**,**b**) and BCG- (**c**,**d**) or BCG-prime and Flu/THSP boost-vaccinated mice (**e**,**f**) after infection with highly virulent *M. tb* H37Rv strain, magnification is 300× and 600×. (**a**) Confluent foci of specific infiltration in nonvaccinated mouse lungs (300×). (**b**) Epithelioid cell granuloma in the foci of specific infiltration in nonvaccinated mouse lungs (600×). (**c**) Foci of specific infiltration in BCG-vaccinated mouse lungs (300×). (**d**) Large granuloma with single epithelioid cells in BCG-vaccinated mouse lungs (600×). (**e**) Small foci of specific infiltration in BCG-prime and Flu/THSP boost vaccinated mouse lungs (300×). (**f**) Small lymphoid granulomas in BCG-prime and Flu/THSP boost vaccinated mouse lungs (600×).

**Figure 5 vaccines-09-01273-f005:**
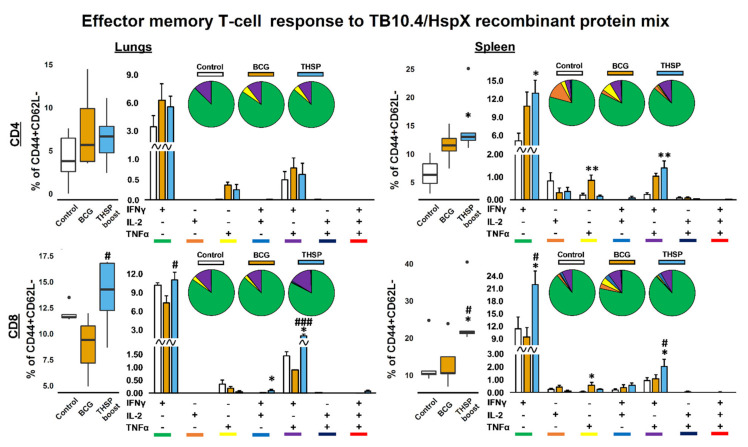
TB10.4/HspX-specific effector memory T-cellular immune response in lungs and spleen of boost-immunized mice, 5 weeks after *M. tuberculosis* challenge. CD4/8^+^CD44^+^CD62L^−^ effector memory (EM) T-cells from lungs and spleen expressing IFNγ, IL-2 and TNFα after 24 h stimulation with TB10.4/HspX recombinant protein mix. Box-plots represent total amount of cytokine-producing EM T-cells. Bar-plots represent the average percent of each cytokine-producing population within total EM T-cells subset (Mean ± SE). Pie-charts show the average percent of each cytokine-producing population within total subset of cytokine-producing cells. Differences between groups were analyzed using Student’s t test. * *p* < 0.05, ** *p* < 0.01, *n* = 5 in comparison with control group; # *p* < 0.05, and ### *p* < 0.001, *n* = 5 compared with BCG.

**Figure 6 vaccines-09-01273-f006:**
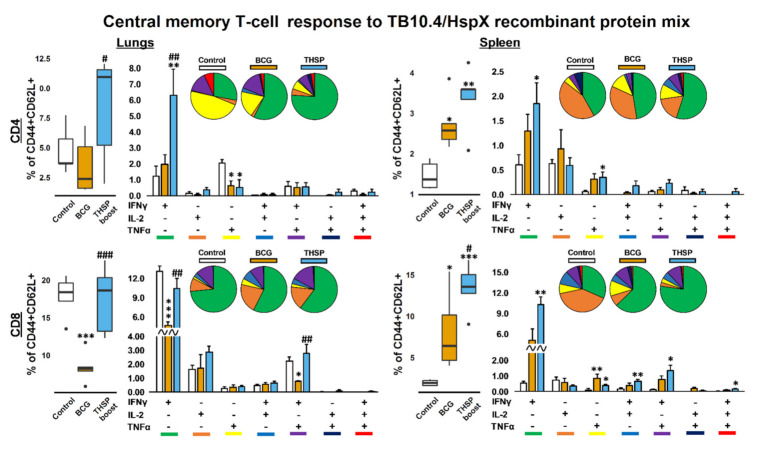
TB10.4/HspX-specific central memory T-cellular immune response in lungs and spleen of boost-immunized mice, 5 weeks after *M. tuberculosis* challenge. CD4/8^+^CD44^+^CD62L^−^ central memory (CM) T-cells from lungs and spleen expressing IFNγ, IL-2 and TNFα after 24 h stimulation with TB10.4/HspX recombinant protein mix. Box-plots represent total amount of cytokine-producing EM T-cells. Bar-plots represent the average percent of each cytokine-producing population within total CM T-cells subset (Mean ± SE). Pie-charts show the average percent of each cytokine-producing population within total subset of cytokine-producing cells. Differences between groups were analyzed using Student’s t test. * *p* < 0.05, ** *p* < 0.01, and *** *p* < 0.001, *n* = 5 in comparison with control group; # *p* < 0.05, ## *p* < 0.01, and ### *p* < 0.001, *n* = 5 compared with BCG.

## Data Availability

The data presented in this study are available on reasonable request from the corresponding author.

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
