# Peer review of "Enhancement of the Local CD8+ T-Cellular Immune Response to Mycobacterium tuberculosis in BCG-Primed Mice after Intranasal Administration of Influenza Vector Vaccine Carrying TB10.4 and HspX Antigens"

_vaccines, 2021, doi:10.3390/vaccines9111273_

Round 1
Reviewer 1 Report
In the manuscript titled “Enhancement of the local CD8+ T-cellular immune response to Mycobacterium tuberculosis in BCG-primed mice after intranasal administration of Influenza Vector Vaccine Carrying TB10.4 and HspX antigens”, the authors described innate and adaptive immune response to immunization with a new mucosal TB vaccine candidate based on recombinant attenuated influenza vector (Flu/THSP) co-expressing TB10.4 and HspX proteins of Mycobacterium tuberculosis as well as the protective efficacy caused by the BCG-prime → Flu/THSP vector boost vaccination in mice. They found that the mucosal administration of Flu/THSP induces the incoming of interstitial macrophages in the lung tissue and stimulates the expression of co-stimulatory CD86 and CD83 molecules on antigen-presenting cells. Their results also claimed that Flu/THSP vaccine protect mice from severe lung injury through inducing the incoming of interstitial macrophages and stimulating the expression of co-stimulatory molecules on APCs and enhancing the CD8+-Tem immune response. Overall, these findings have provided more insights into the Flu/THSP vaccine, but there are still some concerns as following:
Comments:
- M. tuberculosis contains a variety of known antigens, ESAT6, AG85, CFP-10, MTB32, etc., why TB10.4 and HspX were selected as vaccine-specific antigens needs to be discussed.
- The protective mechanism of Flu/THSP vaccine is still lacking. The authors may need to provide the evidence such as adoptive transfer to show what kind of immune cells really contributes to the protective efficacy of Flu/THSP vaccine. The mechanism underlying the induction of innate and adaptive immune responses by TB10.4 and HspX also need to be provided.
- TB10.4 is an important antigen of M. tuberculosis, Yang et al. have shown that those TB10.4 antigen-specific T cells are not able to promote macrophage to eliminate bacteria (PMID: 29782535), then the questions remains whether the enhanced T cell responses by TB10.4 observed in this study can directly promote the clearance of intracellular mycobacteria in macrophages, which can be demonstrated by coculture system of TB10.4 antigen-specific T cells with macrophages.
- In this study, Flu/THSP -boosted mice showed the highest level of CD8+-Tem immune response, but how do TB10.4 and HspX promote the percentage and function of the memory T cell? The detailed underlying mechanism needs to be provided.
Author Response
We are grateful to Reviewer 1 for the critical view of the article, valuable questions and comments.
Q1. M. tuberculosis contains a variety of known antigens, ESAT6, AG85, CFP-10, MTB32, etc., why TB10.4 and HspX were selected as vaccine-specific antigens needs to be discussed.
R1. As it stated in lines 55-59:
“Previously it was shown that TB10.4 serves as a target for antimycobacterial immune responses in BCG-immunized humans and mice. More than 70% of vaccinated or TB-infected individuals react on restimulation with TB10.4 by high levels of IFN-γ production [7]. HspX demonstrates an immunoadjuvant potential, inducing DC maturation and proinflammatory cytokine expression through the activation of MyD88 and TRIF signaling pathways [8].”
It could be also noted, that HspX is demonstrated to induce strong cellular and humoral immune responses in latent phase of tuberculosis infection and is discussed as a perspective candidate vaccine against Mycobacterium tuberculosis. Moreover, TB10.4 and HspX are encoded by Bacillus Calmette-Guerin and, therefore, are useful for a prime-boost immunization scheme.
Additional information is included in the introductory section of the manuscript.
Q2. The protective mechanism of Flu/THSP vaccine is still lacking. The authors may need to provide the evidence such as adoptive transfer to show what kind of immune cells really contributes to the protective efficacy of Flu/THSP vaccine. The mechanism underlying the induction of innate and adaptive immune responses by TB10.4 and HspX also need to be provided.
R2. In the present work, we investigated the immunological properties of our vaccine candidate. The obtained data allow us to claim that the use of the Flu/THSP influenza vector in prime-boost immunization scheme with BCG significantly increases the T-cellular antigen-specific immune response to TB antigens compared to single BCG immunization. We don’t speculate here on the exact mechanisms underlying the observed phenomena, but further investigations will clarify this subject.
Q3. TB10.4 is an important antigen of M. tuberculosis, Yang et al. have shown that those TB10.4 antigen-specific T cells are not able to promote macrophage to eliminate bacteria (PMID: 29782535), then the questions remains whether the enhanced T cell responses by TB10.4 observed in this study can directly promote the clearance of intracellular mycobacteria in macrophages, which can be demonstrated by coculture system of TB10.4 antigen-specific T cells with macrophages.
R3. As it shown on Figure 3, BCG-prime → Flu/THSP boost immunization scheme prevented the inhibition of absorption and digestion capacity of PMF, generally observed in the disseminated tuberculosis infection in mice. This effect could be associated with the direct macrophage-stimulating activity of antigen-specific T-cells or may reflect the nonspecific potentiating action of the influenza vector on the innate immune system. We will address this question in our further studies on this subject.
Q4. In this study, Flu/THSP -boosted mice showed the highest level of CD8+-Tem immune response, but how do TB10.4 and HspX promote the percentage and function of the memory T cell? The detailed underlying mechanism needs to be provided.
R4. The immunogenicity of our Flu/THSP vector is mediated by the self-adjuvanted properties of the NS1-truncated strain. The truncation of NS1 decreases its antagonistic influence on the IFN-I signaling system, provoking a strong innate immune reaction leading to the formation of pronounced Th-1 polarized adaptive immune response, accompanied by the generation of polyfunctional effector CD8+ T-lymphocytes. The immunogenicity and self-adjuvanted potential of NS1-truncated influenza vectors are described in previous publications (Ref. 9-17).
Reviewer 2 Report
This study is a continuation of the earlier data in ref 28, now looking at vaccine responses and T cell response phenotype in immunisation or prime-boost of BL/6 mice. The data offer an increment in terms of T cell response correlates, though falling short of demonstrating mechanism of protection.
Some comments:
- The Intro and abstract offers the reader a slightly quirky view of the current status of TB immunology. There is little discussion of the extensive candidate vaccine landscape (-considerably more extensive than BCG) and the huge dataset on immunogen discovery, far beyond the candidates selected here. As such, the reader has little chance to evaluate these immunogens in context, beyond the credentials proposed in a 2006 citation. Other aspects of the immunology are also a little skewed. One was puzzled by the ref at line 48 about HIV as a Th2 polarised state? This is not a consensus view by any means.
- The study design is clear enough and the findings clear-cut, within the limitations of the BL/6 H37Rv model. The overriding concern is that the attenuated Flu/THSP vaccine appears to have been compared throughout to PBS? If this is indeed the case, it leaves the reader in some doubt as to how much of the lung response phenotype relates to the specific antigens and how much is a response to the flu backbone? If the latter is contributory, as one might surmise, much of the vaccine mechanism becomes a non-specific adjuvant effect. It would be important to see a proper control with an irrelevant antigen insert.
- While the grammar is largely readable, some terms such as 'reviled' (line 183) require attention.
Author Response
We are grateful to Reviewer 3 for the attentive attitude to the article, questions and valuable comments.
Q1. The Intro and abstract offers the reader a slightly quirky view of the current status of TB immunology. There is little discussion of the extensive candidate vaccine landscape (-considerably more extensive than BCG) and the huge dataset on immunogen discovery, far beyond the candidates selected here. As such, the reader has little chance to evaluate these immunogens in context, beyond the credentials proposed in a 2006 citation. Other aspects of the immunology are also a little skewed. One was puzzled by the ref at line 48 about HIV as a Th2 polarised state? This is not a consensus view by any means.
R1. The introductory section is updated with additional information on the current TB vaccines status (Lines 38-41, Lines 53-61)
Q2. The study design is clear enough and the findings clear-cut, within the limitations of the BL/6 H37Rv model. The overriding concern is that the attenuated Flu/THSP vaccine appears to have been compared throughout to PBS? If this is indeed the case, it leaves the reader in some doubt as to how much of the lung response phenotype relates to the specific antigens and how much is a response to the flu backbone? If the latter is contributory, as one might surmise, much of the vaccine mechanism becomes a non-specific adjuvant effect. It would be important to see a proper control with an irrelevant antigen insert.
R2. In previous work, we evaluated the therapeutic effect of vaccination with the Flu/ESAT-6 vectors on the course of established TB infection. A strong synergistic effect was found when vaccination was used in combination with chemotherapy, resulting in a dramatic reduction of bacterial counts in the lungs of infected mice. A partial therapeutic effect was achieved by application of influenza PR8/NS1-125 mutant virus not expressing ESAT-6 in combination with antibiotic therapy which confirms the immunomodulatory activity of influenza vectors themselves (Ref. 25, 28). Thus, the empty vector indeed could provide a certain protective effect as it was shown earlier due to innate immune reaction induced following intranasal immunization (Ref. 26, 27). However, the best protection is archived by the vaccine carrying the specific MTB antigen. Taking this in mind we didn’t focus on the non-specific effects of empty vector in the current study. The main goal of the present work was to compare the protective effects of BCG-prime → Flu/THSP vector boost vaccination with BCG vaccination.
Q3. While the grammar is largely readable, some terms such as 'reviled' (line 183) require attention.
R3. We thank the reviewer for the correction. The detected typos are revised.
Round 2
Reviewer 2 Report
This is somewhat improved